# Perceptions and experiences of women and providers on barriers and facilitators of quality emergency obstetric and newborn care services in public hospitals of West Shoa Zone, Oromia, Ethiopia: A phenomenological qualitative study

Tolera Gudissa Damme[1,2]*, Sileshi Garoma Abeya[3], Dereje Bayissa Demissie[4]

**1** Institute of Health Sciences, Wallaga University, Nekemte, Ethiopia, **2** College of Health Sciences and Referral Hospital, Ambo University, Ambo, Ethiopia, **3** Federal Ministry of Health in Ethiopia, Addis Ababa, Ethiopia, **4** Saint Paul's Hospital Millennium Medical College, Addis Ababa, Ethiopia

* tolerag@gmail.com

## Abstract

### Background

Evidence shows that health care delivery is inadequate and poor quality in low- and middle-income countries. Recently, poor quality of health care has been a more significant challenge for reducing maternal and perinatal mortality than insufficient access to health care services. However, data on barriers and facilitators to quality emergency obstetric and newborn care service in Ethiopia are limited. Therefore, this study aims to explore the perceptions and experiences of women and providers on barriers and facilitators of quality emergency obstetric and newborn care services in public hospitals of West Shoa Zone, Ethiopia.

### Methods

A phenomenological qualitative study was conducted in four selected public hospitals of West Shoa Zone from June 01 to July 30 of 2024. A total of sixty-five study participants were purposively selected and interviewed until data saturation. The data were collected through focus group discussions (38 participants), in-depth interviews (12 participants), and key informant interviews (15 participants). The collected data were transcribed verbatim and translated into English language. Finally, MAXQDA software version 24.4.1 was used for thematic data analysis.

### Results

The study findings were categorized into two major themes (i.e., barriers and facilitators) and three sub-themes (factors related to health care facilities, factors related to health care providers, and factors related to clients and the community

**Data availability statement:** All relevant data are within the paper and its Supporting Information file.

**Funding:** The author(s) received no specific funding for this work.

**Competing interests:** The authors have declared that no competing interests exist.

as a whole). The main identified barriers were a shortage of medical resources and facility infrastructures; insufficient human resources; knowledge and skill gaps of health care providers; inadequate respectful maternity care; a weak client referral system; a lack of updated training; limited community awareness on emergency cases; and inadequate mentoring and supervision activities. In contrast, the major facilitators included the availability of free maternity services, a multidisciplinary team, strong provider commitment and teamwork, establishment of neonatal intensive care units and blood banks, access to ambulances, and community pharmacies.

## Conclusions

This study identified a range of barriers and facilitators influencing the quality of emergency obstetric and newborn care services, encompassing factors related to healthcare facilities, healthcare providers, as well as clients and the whole community. Therefore, all health care facilities and the regional health bureau need to improve the identified barriers and strengthen all facilitators of quality emergency obstetric and newborn care services.

## Introduction

Obstetric emergencies are health problems that are life-threatening for both pregnant women and their newborn babies. They require prompt and focused interventions to save the lives of the mother and/or her babies [1]. Globally, about 260, 000 women died during and following pregnancy and childbirth in 2023, equivalent to over 700 maternal deaths every day, and approximately one every two minutes. Approximately 92% of all maternal deaths occurred in low- and lower-middle-income countries. Sub-Saharan Africa alone accounted for around 70% of maternal deaths (182,000), while southern Asia accounted for around 17% (43,000) [2]. According to preliminary findings from the Ethiopia Demographic and Health Survey (EDHS) of 2024–2025, the the maternal mortality ratio was 141 deaths per 100,000 live births, and the neonatal mortality rate was 25 deaths per 1000 live births [3].

Furthermore, the coverage of institutional deliveries has been significantly increasing over the past few decades. At the same time, a higher proportion of avoidable maternal and perinatal mortality and morbidity has also moved to health facilities, where poor quality care has become a challenge to the quest to end the preventable mortality and morbidity [4]. The research evidences showed that healthcare delivery is inadequate and of poor quality in low- and middle-income countries (LMICs), characterized by issues like shortage of resources, poor infrastructure, lack of political will, and insufficient management and governance, all of which undermine patient safety and health outcomes. The findings of these studies highlight that even when coverage expands, poor quality of care often prevents better outcomes, with factors

like inadequate training, poor adherence to guidelines, and weak health system support contributing to deficiencies in health facility care [5,6].

Globally, more than half of women with obstetric complications do not receive emergency obstetric and newborn care (EmONC) services, with a significant disparity between low (21%), middle (32%), and high-income countries (99%). This disparity corresponds to an annual 11.4 million untreated complications and 951 million women without access to EmONC services [7]. A recent survey on service provision assessment (2021–2022) of EmONC services in Ethiopia among 905 health facilities also reported that only 442 facilities (49%) were able to provide fully functional Basic EmONC (BEmONC) services, while 250 facilities (27.6%) provided fully functional Comprehensive EmONC (CEmONC) services. The national coverage of BEmONC was estimated at 1.5–3.77 facilities per 500,000 population, and CEmONC coverage ranged from 0.83–2.1 facilities per 500,000 populations, indicating remains below the World Health Organization (WHO) recommended standard. The study also revealed significant geographic disparities in service availability, with relatively better access in regions such as Amhara, Southern Nations, Nationalities, and Peoples' Region (SNNPR), and Addis Ababa, while many rural and remote areas had limited access to life-saving obstetric care services [8].

The available evidence reported that nearly half of maternal deaths and approximately one million newborn deaths could be prevented annually through the provision of high-quality care throughout pregnancy, childbirth, and the postnatal period. Nevertheless, in many low-income countries, women receive less than half of the recommended evidence-based maternal health interventions during routine maternity care visits, indicating substantial deficiencies in the quality of care delivered in health facilities (12, 13). The quality of care requires availing skilled birth attendants for every pregnant woman and their newborn and evidence-based care, including respectful care, creating supportive environments, use of effective clinical and non-clinical interventions, health care infrastructure capacitation, health care providers' skills, and positive attitude building toward the provision of care [9].

Furthermore, in most sub-Suharian Africa (SSA) countries, where many health facilities lack some necessary medical equipment and essential drugs [10], the quality of EmONC services are not adequate [11]. A shortage of beds and a lack of separate rooms for different maternity services were frequently reported factors affecting the quality of EmOC [10,12]. The available healthcare providers are inadequate in number, and most of them lack essential skills due to insufficient training [13]. The behavior of healthcare providers usually affects the quality of care provided, particularly when the providers who have a negative attitude are treating the clients [14]. Some studies also reported that factors related to poor management of EmONC provision, including lack of supervision [15,16], delayed patient referral [10,17], and poor staff motivation [18], are challenging the provision of appropriate obstetric care services.

The period around childbirth is the most critical for saving the maximum number of maternal and newborn lives and preventing stillbirths. WHO sees a future in which "Every pregnant woman and newborn receives high-quality care throughout pregnancy, childbirth and the postnatal period" [14]. But, a conducted study indicated that poor quality of facility-based care for women and newborns is the major contributing factor to the increased rates of maternal and perinatal mortality and morbidity [13]. Advocating for the highest attainable quality of care for every mother and newborn is an imperative to achieve the SDGs since high maternal and neonatal mortality are attributable to poor quality of care [19].

But, to the knowledge of the researchers, little is known about the barriers and facilitators of quality emergency obstetric and newborn care services in Ethiopia, specifically in the study area. Additionally, factors affecting the quality of emergency obstetric and newborn care services are often inadequately captured in quantitative studies. A more comprehensive understanding of these factors can be achieved through conducting the qualitative research approaches. Therefore, this study aims to explore the perceptions and experiences of women and providers on barriers and facilitators of quality emergency obstetric and newborn care in public hospitals of west shoa zone, Central Ethiopia. Finally, West Shoa Zone was selected due to its large and diverse population, which is considered representative of many parts of the region in terms of demographic characteristics, health service structure, and maternal and newborn care practices.

## Methods

### Study setting and period

This study was conducted in four public hospitals of the West Shoa Zone: Ambo University Referral Hospital, Ambo General Hospital, Gedo General Hospital and Ginchi Primary Hospital. The four hospitals were selected purposively based on service availability, service capacity, patient volume, and representation of different levels of healthcare delivery within the West Shoa Zone. The West Shewa Zone is found at a distance of 115 KM away from Addis Ababa in the west direction. The West Shewa Zone has twenty-two districts and one town administration, Ambo town. According to the report of West Shoa Zone Health Office, there were a total of 2,821,589 people living in west shoa zone in the area of 14,788.78km². From a total population, about 97,909 were expected pregnant women in the zone in 2022/2023GC. In the West Shoa Zone, there are a total of 09 public hospitals, 92 health centers, and 529 health posts. The emergency obstetric care services are mainly provided by health professionals such as midwives, general medical doctors, integrated emergency obstetric surgeons, and gynecologists [20]. The study period was from June 01 to July 30 of 2024.

### Study design

A phenomenological qualitative study design was used to explore the barriers and facilitators of EmONC services by gathering data from multiple potential sources. This approach was chosen because it is an appropriate design and well suited to exploring individuals' lived experiences and perceptions of a phenomenon. Since the quality of care is influenced by personal experiences, interactions, and contextual factors within health facilities, a phenomenological approach allows for an in-depth understanding of these experiences and the meanings participants attach to them [21]. We used the Standards for Reporting Qualitative Research (SRQR) guidelines when preparing the manuscript to ensure clarity, rigor, and transparency throughout the research process.

### Study population

The study population for this study was all purposively selected women treated for obstetric emergencies (preeclampsia/eclampsia, antepartum hemorrhage, obstructed labor, and postpartum hemorrhage cases); healthcare workers (gynecologists, integrated emergency surgical officers, nurses and midwives); and different levels of facility managers (maternity department heads, neonatal intensive care unit department heads, obstetric and gynecology department heads, nurse directors, medical directors, and hospital chief executive officers) in selected public hospitals of the West Shoa Zone.

### Sampling methods

As to the sampling method, four public hospitals (Ginchi Hospital, Gedo Hospital, Ambo General Hospital, and Ambo University Referral Hospital) were selected purposively from the total nine hospitals found in West Shoa Zone based on their obstetric caseloads and resource limitations. Then, different potential study populations were included through purposive sampling using multiple data collection methods until data saturation. Data saturation is when new data consistently repeats the previous information and codes. Finally, four focus group discussions (FGD) with thirty-eight obstetric care providers, twelve in-depth interviews (IDI) with women treated for obstetric emergencies; and fifteen key informant interviews (KII) with different levels of facility managers were conducted to address the objective of this study.

### Data collection instruments and methods

The open-ended interview guides of data collection instruments were adapted from previous similar literature [22–26] and used for the data collection purposes as the file attached below (S1 File). The facility head assisted in the recruitment of the study participants to obtain rich information about the phenomenon. All interviews and discussions were conducted in a separate quiet room to avoid interruption from outside and to maintain the privacy. The free flow of information was

encouraged through probing. All interviews were tape-recorded with the consent of interviewees. A unique identification number was assigned to all the recorded files. A total of four data collectors having previous experiences of qualitative data collection were involved in the data collection activities. The researchers also considered how their backgrounds and perspectives might influence the study and used reflexive practices throughout data collection process to enhance transparency and reduce bias.

## Trustworthiness of the study/ study rigors

Initially, the investigators prepared the study protocol, including the aim, study design, data collection methods, data analysis, and reporting. Based on the protocol of the activity plan, careful sampling and participant selection for FGDs, IDIs, and KIIs were performed to provide sufficient information about the research context and to assure the transferability of the study findings. The study tool was translated into the local language for the interview. The principal investigators and research assistants were also fluent speakers of the local language and familiar with prior experience with qualitative research and adequate academic background to conduct the data collection. Additionally, Audio records of study participants' interviews, notes taken during the interviews, and transcriptions were saved for cross-checking the process and to sustain the consistency of the interpretation. This study also included an audit of written transcripts, field notes, coding and analysis, interpretation, and other documents to ensure data rigors. To reduce bias during data analysis and the interpretation of the results, the respondents' expressions were recorded. Finally, the overall trustworthiness of the data was ensured through the systematic application of specific strategies that addressed data credibility, transferability, dependability, and confirmability. These strategies included the use of multiple data sources, maintaining detailed documentation of all research data, clearly describing participant characteristics, and supporting interpretations with verbatim quotes.

## Data processing and analysis

The collected data was transcribed verbatim into Afan Oromo (the regional working language) by listening to the tapes recorded again and again. The transcripts were subsequently translated to the English language with additional cross-checking of audio records and the notes taken during the interview conducted prior to the data analysis. Then, the data were first saved in plain text format and imported into MAXQDA software version 24.4.1 to facilitate coding and categorizing for data analysis. Finally, an inductive thematic analysis was applied, and the findings of the study were presented with texts and tables.

## Ethical clearance and consent to participate

The ethical clearance was obtained from the ethical review board of Wallaga University, School of Postgraduate with a referance (IRB reference No.WU/RD/718). Then, the ethical clearance letter was given to the concerned bodies of selected hospitals after providing information about the purposes of the study. Additionally, the benefits of the study were explained to the participants, and a written informed consent was obtained prior to the interviews. Finally, all the study participants were informed that their participation was voluntary and they had the right to withdraw from the study at any stage of data collection.

## Result

### Description of the study participants

The study involved a total of sixty-five study participants, including twelve in in-depth interviews with women receiving EmONC services, thirty-eight providers in four focused group discussions, and fifteen key informant interviews with different levels of health facility managers at four public hospitals in the West Shoa Zone.

The in-depth interview was conducted among twelve purposively selected women treated for obstetric emergencies including preeclampsia/eclampsia, antepartum hemorrhage, obstructed labor, and postpartum hemorrhage cases and the socio-demographic characteristics of the included study participants were described as below (Table 1).

Additionally, the focus group discussions (FDGs) were also conducted among thirty-eight obstetric care providers (including midwives, clinical nurses, neonatal nurses, and integrated emergency surgical officers (IESO) professionals) and the socio-demographic characteristics of the included study participants were described as below (Table 2).

Furthermore, fifteen key informant interviews (KIIs) were also conducted with different levels of health facility managers (including the Maternity department head, Neonatal intensive care unit department head, Obstetric and Gynecology department head, Nurse director/Matron, Medical director and the Hospital Chief Executive Officer and the

**Table 1. The socio-demographic characteristics of participants in in-depth interviews conducted in public hospitals of the West Shoa Zone, Central Ethiopia, 2024.**

| 1. Socio-demographic characteristics of in-depth interview study participants | | | |
|---|---|---|---|
| **Study Variables** | | **Number** | **Percent** |
| Age | < 30 years | 9 | 75.0 |
| | ≥ 30 years | 3 | 25.0 |
| Educational status | Primary education | 3 | 25.0 |
| | Secondary education | 5 | 41.7 |
| | College and above | 4 | 33.3 |
| Marital status | Married | 12 | 100.0 |
| Occupational status | Farmers | 4 | 33.3 |
| | Gov't workers | 3 | 25.0 |
| | Merchants | 2 | 16.7 |
| | House wife | 3 | 25.0 |
| Mother case of admissions | Severe pre-eclampsia/Eclampsia | 3 | 25.0 |
| | Antepartum hemorrhage | 3 | 25.0 |
| | Obstructed labor | 3 | 25.0 |
| | Post-partum hemorrhage | 3 | 25.0 |

**Table 2. The socio-demographic characteristics of participants in focus group discussions conducted in public hospitals of the West Shoa Zone, Central Ethiopia, 2024.**

| 1. Socio-demographic characteristics of focus group discussion study participants | | | |
|---|---|---|---|
| **Study Variables** | | **Number** | **Percent** |
| Age | <30 Years | 26 | 68.4 |
| | ≥30 Years | 12 | 31.6 |
| Sex | Male | 15 | 39.5 |
| | Female | 23 | 60.5 |
| Educational status | Bsc | 37 | 97.4 |
| | Msc | 1 | 2.6 |
| Category of professional | Midwives | 19 | 50.0 |
| | Nurses | 18 | 47.4 |
| | Integrated Emergency Surgical Officers | 1 | 2.6 |
| Experiances of work | < 5 Years | 11 | 28.9 |
| | 5-10 Years | 21 | 55.3 |
| | > 10 years | 6 | 15.8 |

socio-demographic characteristics of study participants included in the key informant interviews were described as below (Table 3).

## Barriers and facilitators of quality emergency obstetric care services

This study found that a wide range of barriers and facilitators of quality emergency obstetric care services were identified, and the study findings were presented in the following two major research objectives (i.e., barriers of quality emergency obstetric care services and facilitators of quality emergency obstetric care services).

## Themes identified on barriers to quality of emergency obstetric care services

**Barriers related to health care facilities.** The first category of health facility-related barriers was related to shortages of medications and medical supplies, especially in the maternity room, operating theater, and neonatal intensive care unit. Facility infrastructure such as a shortage of rooms, patient beds, operating theater tables, water supply, and functional latrines in the maternity ward was also another health facility-related barrier to providing quality emergency obstetric care services. Some laboratory services such as chemistry tests were also sometimes absent from the facility.The following are what some of the study participants reported during the data collection period:

One KII study participant reported the facility related barriers to providing quality of emergency obstetric care services as follows:

………" With regard to resource availability, most health facilities experience a shortage of medical supplies. Some organ function tests related to laboratory services were occasionally not performed due to interruptions in the supply of reagents. The laundry machine also doesn't work sometimes, and there was an interruption of cesarean section (C/S)

**Table 3. The Socio-demographic characteristics of participants in key informant interviews conducted in public hospitals of the West Shoa Zone, Central Ethiopia, 2024.**

| 2. Socio-demographic characteristics of key informant interview study participants | | | |
|---|---|---|---|
| **Variables** | | **Number** | **Percent** |
| Age | <30 Years | 6 | 40.0 |
| | >30 Years | 9 | 60.0 |
| Sex | Male | 13 | 86.7 |
| | Female | 2 | 13.3 |
| Educational status | Msc | 3 | 20.0 |
| | Bsc | 12 | 80.0 |
| Professions | Midwives | 6 | 40.0 |
| | Clinical Nurse | 7 | 46.7 |
| | General practitioner | 2 | 13.3 |
| Work Experiences | <5 Years | 3 | 20.0 |
| | 5- 10 Years | 10 | 66.7 |
| | >10 Years | 2 | 13.3 |
| Position/role in the facility | Medical director | 2 | 13.3 |
| | Nurse director/Mitron | 4 | 26.7 |
| | Maternity department head | 4 | 26.7 |
| | Obstetric and Gynecology department head | 3 | 20.0 |
| | Nneonatal intensive care unit department head | 2 | 13.3 |
| Duration in the position | < 1 Year | 3 | 20.0 |
| | 1-5 Years | 12 | 80.0 |

services for some duration. But we were taking laundry clothes to the nearby health facilities for laundry service, and the problem was solved after a few periods...……… (Age: 31; Profession: Midwife; Experience: 8 years; KII Participant No. 01).

For instance, another IDI study participant explained the problem by narrating:

…....……. "There is a shortage of drugs in the facility, and clients buy the drugs from outside. There is also a shortage of laboratory investigations, and some investigations were done out of the facility. Therefore, there is a delay of care due to the investigations sent out of the health facility" …….. (Age: 32; IDI Participant No. 06).

In addition, the FGD discussants described health facility-related barriers for providing quality of emergency obstetric care services as follows:

………" When we look the hospital infrastructure, there were not enough rooms and beds in the maternity department. There was also no separate latrine for clients and health care providers. The hygiene of the available latrine was also very bad. Related to the C/S services, there was unnecessary referral of clients to the other hospitals due to a shortage of blood for transfusion. The general surgery and obstetric surgery were also using one operating table, and this caused a delay in providing emergency obstetric care management……….." (Age: 29; Profession: Midwife; Experience: 8 years; FGD-3 Participant No. 03).

Another FGD participant explained the problem:…………"When we look at the neonatal intensive care unit (NICU) department, the room is not arranged as standard to manage the neonatal complications. The infectious and non-infectious room should be separated, but we used one room for all neonates. The kangaroo mother care (KMC) was also not arranged to be provided in the NICU department. The other barriers were the lack of medical equipment, such as a heater (only one heater available), an incubator, and the like. The hospital water supply and electric light were having a gap in the NICU department, and it needed to be corrected for providing quality emergency obstetric and newbrn care services"……… (Age: 30; Profession: Nurse; Experience: 11 years; FGD-3 Participant No. 02).

The other facility-related barriers for providing quality of emergency obstetric care services were the client referral system and the functionality of the liaison office, as most client referrals were done without communication with the liaison office, and the liaison office of some health facilities was not functional. Absence of feedback on client referral linkage, unwillingness of some facilities to take the referred clients, and unnecessary referral of some clients were also reported as facility-related barriers to provide quality of emergency obstetric care services.

One KII study participant reported: ……………"One of the facility-related barriers to provide quality emergency obstetric care is the problem of referral linkages among health facilities. There is no communication among health facilities when they referred clients, and they were referred from far areas without communication. The healthcare facility liaison office was not working appropriately, and this made a delay to getting care and needs to be corrected to improve the quality of emergency obstetric care services"……….. (Age: 38; Profession: Midwife; Experience: 20 years; KII Participant No. 02).

Furthermore, another facility-related barrier was the shortage of health care professionals as standard, especially in the operating theater room and the neonatal intensive care unit. The available healthcare providers have also not received updated EmONC training. There was also a high client caseload at the facility and a scarcity of ambulances for client referral. These barriers may lead the mother and their child to encounter a delay of care and other obstetric risks. Related to the supervision and mentoring activity, previously there was a mentoring activity from a higher health facility. But currently, there is no mentoring activity except some internal supervision by facility management. Fewer management commitments and the absence of obstetric guidelines were also other barriers to providing quality emergency obstetric and newborn care services.

The key informant interview participant reported: ……………… "There are several facility-related barriers, including a shortage of essential human resources, challenges in the client referral system, a shortage of ambulances, and insufficient EmONC training for healthcare providers. The number of staff providing emergency obstetric care services does not meet facility standards, particularly in the neonatal intensive care unit and operating theater departments. There are

issues with the referral system, as mothers are referred without proper liaison communication, and the health facility faces a shortage of ambulances. Previously we had two ambulances, but currently only one ambulance is functional. Because, when one ambulance goes with a referral to another facility, another mother may also need to be referred. Furthermore, previously there was a training of updated EmONC services, but currently there is no such training due to unknown reasons" …………. (Age: 31; Profession: Midwife; Experience: 8 years; KII Participant No. 01).

One FGD participant of a nurse profession working in neonatal intensive care unit also reported: ………."Since we are in the African countries, the needed resources and human powers for providing quality emergency obstetric care services are not fulfilled as standard. As a standard, the neonatal intensive care unit needs to have its own laboratory facility, but in our case there is no laboratory facility in the neonatal intensive care unit. The category of health professionals working in the neonatal intensive care unit is not as standard according to their specialization. Another problem is the lack of updated training, including neonatal resuscitation. So, health care providers need to get the updated training to provide quality emergency obstetric care services" …………. (Age: 26; Profession: Nurse; Experience: 8 years; FGD – 4 Participant No. 01).

**Barriers related to health care providers.** The main barriers related to healthcare providers included gaps in knowledge, skills, and motivation to deliver quality emergency obstetric care, which may lead to delays in receiving such care and increase the risk of adverse outcomes for mothers. Additionally, compassionate and respectful maternity care and client counseling services were not adequately provided by healthcare providers due to the gaps in knowledge, skills, and motivation or due to the high client caseload in the health care facilities.

One KII study respondent explained the issues:………."Barriers related to health care providers to deliver quality emergency obstetric care services are influenced by the knowledge and skills of healthcare providers, their motivation and commitment, and problems related to insufficient incentives for different providers" ………… (Age: 32; Profession: Medical Doctor; Experience: 6 years; KII Participant No. 13)

Another study participant involved in the FGD reported: …………"Staff motivation to provide quality health services is decreased due to various factors, such as issues related to salary, workload, risk allowances, and the lack of health insurance coverage for healthcare providers" …….……… (Age: 26; Profession: Nurse; Experience: 8 years; FGD – 4 Participant No. 01).

The other IDI participant witnessed the problem ……….. "Related to the counseling given by health care providers, adequate information and counseling was not provided for all obstetric mothers and their families. Therefore, full counseling about the obstetric emergencies and the C/S procedure should be given for the obstetric mothers and their families, and the community as whole" …….…….. (Age: 25; IDI Participant No. 11).

**Barriers related to clients and the community as whole.** The main barriers related to clients and the community as whole is a gap of awareness about obstetric emergency cases and the need for immediate obstetric care. This information gap leads to delays in taking emergency obstetric care services and other obstetric risk factors.

One KII participant reported …….."one of the client related barriers to quality care was some clients perceived that mothers who delivered by C/S never get another child later, and they didn't take the clinician decision of C/S delivery for obstetric management. This makes a delay to care until convincing the mother and/or their families. Some clients also delay to EmONC services due to long distance, lack of transportation, and security issues" ….........(Age: 32; Profession: Medical Doctor; Experience: 6 years; KII Participant No. 13).

### Themes identified on facilitators of quality emergency obstetric care services

**Facilitators related to health care facilities.** The major facilitators related to healthcare facilities were the availability of free maternity care services and the availability of a multidisciplinary team in the facility, including obstetric and gynecology specialists, integrated emergency surgery officers, and midwives in the facility. Furthermore, other healthcare facility–related facilitators included the establishment of neonatal intensive care units and blood banks, the availability of

liaison offices for client referrals, and the presence of ambulances and community pharmacies within public healthcare facilities. The provision of food and obstetric services every 24 hours was also identified as a key factor that facilitates quality emergency obstetric care.

One FGD participant said:.........."What I think as a facilitator for quality of emergency obstetric care services is there are human resources with multidispilinary team, who are serving the community, and there are four ambulances in the facility for emergency referral cases"........... (Age: 26, Profession: Nurse; Experience: 6 years; FGD-2 Participant No. 08).

Another KII participant explained……...……...….. "Obstetric care services are provided free of charge in our facility. There is also an ambulance service for referral cases, as well as a functional liaison office that facilitates referral linkages for emergency clients" …………… (Age: 32; Profession: Midwife; Experience: 8 years; KII Participant No. 11).

**Facilitators related to health care providers.** The main facilitators related to health care providers were the providers' good commitment and teamwork for patient care and the providers' respectful maternity care. A midwife working at one hospital said:

............................. "What we see as facilitators in our facility is the health care workers are working with a good commitment, using the available resources to save the lives of mothers with all their all capacity and energy. Respectful maternity care and good communications among health care providers were also another facilitators for good quality care" …………..…….. (Age: 28; Profession: Midwife; experience: 5 years; FGD-3 participant No. 07).

**Facilitators related to clients and the community as whole.** The major facilitators related to clients and the community as whole is the community willingness to get obstetric services from health care providers in the health care facilities and the community support provided for mothers' transportation to the health facility during emergency cases for immediate emergency obstetric care services.

One KII study participant saying: ………….."When a pregnant woman encounters obstetric complications, the community, being aware of the dangers of such complications, takes her to the nearest health facility as soon as possible to save the lives of both the mother and the baby ……………..….." (Age: 25; Profession: Midwife; Experience: 4 years; KII – Participant No. 05).

In general, each category of the identified barriers and facilitators of quality emergency obstetric care services in public hospitals of west shoa zone, central Ethiopia had several sub-themes, as illustrated in the table below (S2 File).

## Discussion

This study explored the perceptions and experiences of women and providers on barriers and facilitators of quality emergency obstetric and newborn care services in public hospitals of the West Shoa Zone, Central Ethiopia. The study findings reported that several barriers and facilitators of quality emergency obstetric care services were identified as themes, such as factors related to health care facilities, factors related to health care providers, and factors related to the client and community as a whole.

The present study identified critical facility-related barriers to the provision of quality emergency obstetric and newborn care, particularly shortages of medications, medical equipment, and essential supplies. Additionally, health facility infrastructures such as scarcity of rooms, patient beds, operating tables, lack of water supply in the maternity ward, and poor latrine facilities were also reported as barriers to quality of emergency obstetric and newborn care services. This study finding was supported by previous studies conducted in Bangladesh [27], Ghana [28], Malawi [29], Gambia [30], and the different parts of Ethiopia [22–26,31], suggesting that such constraints are widespread and persistent across low-resource settings. The lack of these essential health facility resources can significantly hinder the quality of emergency obstetric care, potentially leading to preventable maternal and neonatal morbidity and mortality.

Another study conducted in the Philippines to identify factors influencing the provision of basic emergency obstetric care services reported four key themes as barriers to quality care: infrastructure, human resources, referral systems, and local government support. Factors and barriers to each sub-theme were inadequate infrastructures, non-functioning

facilities, lack of essential equipment and supplies, poor data management, lack of transport, poor communication channels, lack of capacity-building opportunities, and lack of monitoring and suppervision [32]. This finding indicates that inadequate infrastructures, limited essential supplies, weak referral and communication systems, and gaps in provider capacity can delay timely interventions and compromise the quality of emergency obstetric and newborn care.

This study also reported that facility-related barriers such as high client caseloads, shortage of ambulance/transportation, lack of training, and absence of supportive supervision were the barriers to providing quality of EmONC services. This finding was similar to studies conducted in Addis Ababa, Ethiopia, which indicated lack of transportation, overcrowding at referral hospitals, insufficient training, and absence of supportive supervision were the barriers to provision of quality EmONC services [33,34]. These findings indicate that facility-related barriers can hinder life-saving interventions and limit healthcare professionals' ability to appropriately manage the obstetric emergencies. Furthermore, the facility-level constraints can reduce the efficiency, safety, and responsiveness of emergency obstetric care, ultimately affecting the maternal and newborn health outcomes.

The study also identified barriers related to health care providers, including knowledge and skills gaps of HCPs, less motivation and commitment of health care providers, poor respectful maternity care, shortage of HCPs, lack of updated training on EmONC services, and high turnover of skilled HCPs. These barriers can lead to reduced adherence to the standards of care, delayed or inappropriate management of obstetric emergencies, and increased risk of preventable maternal and neonatal mortality and morbidity. This finding is consistent with a study conducted in the Tigray region, which reported that poor quality of care during childbirth and postpartum was associated with gaps in providers' skills and knowledge, inadequate motivation schemes, and suboptimal caring behaviors among providers [26].

The study reported that a gap of awareness about obstetric emergency cases and delays of emergency obstetric care services were barriers related to the client and community for receiving quality emergency obstetric and newborn care services. Similar findings were reported in Malawi that indicated the client-related factors contributing to poor quality of care included delays in seeking care and a lack of awareness regarding the signs of obstetric emergencies [29].

Another study conducted in the Wolaita Zone of Southern Ethiopia also revealed that service users' perceptions and experiences (knowledge, perceived quality, reputation, respectful care, and gender) and community-related factors (misconceptions, traditional practices, and family and peer influence) were barriers to emergency obstetric and newborn care services [22]. The findings highlight that low community awareness of obstetric danger signs and delays in seeking care can hinder timely access to emergency obstetric and newborn care and increase the risk of preventable maternal and neonatal morbidity and mortality.

This study identified the main health facility-related facilitators of quality emergency obstetric and newborn care services as including the availability of free maternity care, a multidisciplinary team within the facility, availability of liaison office for client referrals, established blood banks and NICU services, availability of ambulances and community pharmacies at public health care facilities, and continuous 24-hour obstetric care services. These factors collectively enhance the readiness and capacity of health facilities to respond effectively to obstetric emergencies and improve the maternal and neonatal outcomes. This finding was supported by previous studies conducted in Ethiopia [22,23], which highlighted that the combination of well-resourced facilities, competent healthcare providers, and effective health system organization are the key facilitators of quality EmONC services in sub-Saharan Africa.

The healthcare provider–related facilitators of quality emergency obstetric and newborn care services were providers' commitment and teamwork, respectful maternity care, and effective communication among healthcare providers. This finding is consistent with the WHO quality of care framework, which emphasizes respectful care and effective communication as essential components of high-quality maternal and newborn services [35]. It is also supported by a study on facility readiness and the experiences of women and health care providers in receiving and delivering obstetric care at comprehensive health posts in Ethiopia found that women reported positive experiences, particularly highlighting compassionate care and improved access due to the proximity of health care providers as the main factors

facilitating quality emergency obstetric and newborn care services [25]. Providers' commitment, respectful maternity care, and effective communication are clinically important because they ensure timely, safe, and patient-centered emergency obstetric and newborn care.

The study reported that community willingness to get obstetric care from health care providers, community awareness on pregnancy danger signs, and community support for mothers' transportation during emergency cases were the main facilitators of community related factors for taking quality emergency obstetric and newborn care services. These findings are consistent with another Ethiopian study indicating that improved awareness and behavioral shifts at the individual and community levels are key facilitators of maternal and newborn health service utilization [23]. This implies that community-level factors play a critical role in determining whether women receive high-quality emergency obstetric and newborn care. Therefore, improving clinical outcomes is not only dependent on facility readiness, but also on how well the community recognizes and responds to obstetric emergencies.

## Strength and limitations of the study

A key strength of this study is its comprehensive examination of the experiences of both women and healthcare providers, identifying barriers and facilitators that affect the quality of emergency obstetric care services in public hospitals of West Shoa Zone.The study also provides in-depth insights into the perceived barriers and facilitators of quality emergency obstetric care in the study setting, which will be helpful to shape the holistic programming of quality emergency obstetric care services. But, as limitations, the selection bias may occur, as participants were purposively recruited from specific public hospitals, which may not reflect the experiences of all women and healthcare providers in the region. The recall bias was also possible, particularly for participants describing past experiences, and the data were based on self-reported accounts, which may be influenced by social desirability. Furthermore, as with most qualitative research, the findings are context-specific and may have limited generalizability to other healthcare settings or regions with different cultural and health system contexts.

## Conclusion and recommendations

The study identified several barriers to the quality of EmONC services: shortage of medications, medical equipment, and medical supplies; laboratory investigations; facility infrastructures; delay in obstetric care; poor patient referral system; shortage of human resources; lack of capacity-building training; knowledge and skill gap of health care providers; and insufficient mentoring and supervision activities for obstetric care services. The identified facilitators were availability of free maternity services, availability of ambulances and multidisciplinary team of health care providers', establishment of blood banks and NICU services, providers' commitment and teamwork, and availability of ambulances and community pharmacies at public health facility. Therefore, the health facilities need to work harder to secure the resources required to improve the quality of care provided for all obstetric mothers. Additionally, the regional health bureau must ensure that all healthcare facilities are adequately equipped with necessary resources including medications, medical supplies, infrastructure, and human resources by actively involving all stakeholders in the health sector. Furthermore, addressing the identified barriers and strengthening the facilitators of quality services are very essential for enhancing the quality of EmONC services.

## Supporting information

**S1 File. The English version of guiding questionaries prepared for data collection process.**
(PDF)

**S2 File. A summary results on barriers and facilitators of quality EmONC services.**
(PDF)

## Acknowledgments

We acknowledge Wallaga University, Ambo University, study participants, data collectors and facility managers at the study conducted.

## Author contributions

**Conceptualization:** Tolera Gudissa Damme.

**Formal analysis:** Tolera Gudissa Damme.

**Investigation:** Tolera Gudissa Damme.

**Methodology:** Tolera Gudissa Damme, Sileshi Garoma Abeya, Dereje Bayissa Demissie.

**Supervision:** Sileshi Garoma Abeya, Dereje Bayissa Demissie.

**Validation:** Sileshi Garoma Abeya, Dereje Bayissa Demissie.

**Visualization:** Sileshi Garoma Abeya, Dereje Bayissa Demissie.

**Writing – original draft:** Tolera Gudissa Damme.

**Writing – review & editing:** Sileshi Garoma Abeya, Dereje Bayissa Demissie.

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
