## [Decision Letter · Decision Letter 0]

25 Mar 2025

PONE-D-25-03007PERCEPTIONS AND EXPERIENCES OF WOMEN RECEIVING EMERGENCY OBSTETRIC CARE SERVICES AND OBSTETRIC CARE PROVIDERS ON BARRIERS AND FACILITATORS OF QUALITY CARE AT PUBLIC HOSPITALS IN OROMIA, ETHIOPIA: A PHENOMENOLOGICAL QUALITATIVE STUDY DESIGNPLOS ONE

Dear Dr. Damme,

Thank you for submitting your manuscript to PLOS ONE. After careful consideration, we feel that it has merit but does not fully meet PLOS ONE’s publication criteria as it currently stands. Therefore, we invite you to submit a revised version of the manuscript that addresses the points raised during the review process.

We look forward to receiving your revised manuscript.

Kind regards,

Dinaol Abdissa Fufa, Mph

Academic Editor

PLOS ONE

3. Thank you for stating the following in your Competing Interests section: [No].

6. Please amend the manuscript submission data (via Edit Submission) to include authors Dr. Sileshi Garoma Abaya and Dereje Bayissa Demissie.

Reviewers' comments:

Reviewer's Responses to Questions

**Comments to the Author**

1. Is the manuscript technically sound, and do the data support the conclusions?

Reviewer #1: Yes

Reviewer #2: Partly

2. Has the statistical analysis been performed appropriately and rigorously? 

Reviewer #1: No

Reviewer #2: N/A

3. Have the authors made all data underlying the findings in their manuscript fully available?

Reviewer #1: Yes

Reviewer #2: Yes

4. Is the manuscript presented in an intelligible fashion and written in standard English?

Reviewer #1: Yes

Reviewer #2: No

5. Review Comments to the Author

Reviewer #1: This article provides an overview about the perceptions and experiences of women treated for obstetric emergencies and their obstetric care providers on barriers and facilitators of quality emergency obstetric care services, which will provide an insight for policy makers to protect maternal morbidity and mortality. I would give the following comments to the authors

1. In the introduction part, you are trying to identify the gap of your study; it will be more prices if you support it by reference.

2. In the methodology part, you write that there are 9 public hospitals in the zone, but you gather data from 4 hospitals. By what method do you select those 4 hospitals?

3. Your sampling method and sample size determination was not clearly written. Write it clearly the sampling technique and sample size.

4. On your result page 14, the total participants were 63. But on your sample determination the participants were: 4 FGD groups with obstetric care providers, 10 IDI with mothers and 15 KII with facilitators. So does 63=29? Also in table 1 the total number of IDI are 12 but it is 10 in sampling method.

5. Both of the graphs are not readable.

Reviewer #2: The authors have chosen a very relevant topic of determining the facilitators and barriers to the quality of EMNOC services in a LMIC, with regards to its importance in preventing maternal and perinatal morbidity and mortality.

The methodology chosen is appropriate in order to explore, in depth, the enablers and barriers of this service. The subjects chosen are appropriate and include patients , HCPs and Key informants.

However, the English Language used needs to be properly edited in order to be intelligible.

The results have mentioned the barriers to provision of EMNOC services. However the facilitators mentioned are also similar to the barriers. e.g Barriers mentioned are unavailability of proper referral system, ambulances, trained staff, medical supplies etc. However, the faciliatators mention availability of all above factors. This needs to be clarified in discussion. Discussion also needs to provide some solutions to the perceived barriers in order to improve the quality of EMNOC provision.

6. PLOS authors have the option to publish the peer review history of their article (what does this mean?). If published, this will include your full peer review and any attached files.

Reviewer #1: No

Reviewer #2: No

---

## [Author Response · Author response to Decision Letter 1]

3 Apr 2025

The response for reviewer and editor comments were attached as independent file with the file name " Response to reviewers".

---

## [Decision Letter · Decision Letter 1]

22 Oct 2025

PONE-D-25-03007R1PERCEPTIONS AND EXPERIENCES OF WOMEN RECEIVING EMERGENCY OBSTETRIC CARE SERVICES AND OBSTETRIC CARE PROVIDERS ON BARRIERS AND FACILITATORS OF QUALITY CARE AT PUBLIC HOSPITALS IN OROMIA, ETHIOPIA: A PHENOMENOLOGICAL QUALITATIVE STUDY DESIGNPLOS ONE

Dear Dr. Damme,

Thank you for submitting your manuscript to PLOS ONE. After careful consideration, we feel that it has merit but does not fully meet PLOS ONE’s publication criteria as it currently stands. Therefore, we invite you to submit a revised version of the manuscript that addresses the points raised during the review process. Kindly ensure that all reviewer comments are thoroughly addressed and that the resubmission strictly adheres to the journal’s submission guidelines.  Please submit your revised manuscript by Dec 06 2025 11:59PM. If you will need more time than this to complete your revisions, please reply to this message or contact the journal office at plosone@plos.org. Please include the following items when submitting your revised manuscript:

We look forward to receiving your revised manuscript.

Kind regards,

Addis Eyeberu

Academic Editor

PLOS ONE

Journal Requirements:

Additional Editor Comments:

Kindly ensure that all reviewer comments are thoroughly addressed and that the resubmission strictly adheres to the journal’s submission guidelines.

Reviewers' comments:

Reviewer's Responses to Questions

**Comments to the Author**

1. If the authors have adequately addressed your comments raised in a previous round of review and you feel that this manuscript is now acceptable for publication, you may indicate that here to bypass the “Comments to the Author” section, enter your conflict of interest statement in the “Confidential to Editor” section, and submit your "Accept" recommendation.

Reviewer #1: All comments have been addressed

Reviewer #3: (No Response)

Reviewer #4: (No Response)

2. Is the manuscript technically sound, and do the data support the conclusions?

Reviewer #1: Yes

Reviewer #3: Partly

Reviewer #4: Partly

3. Has the statistical analysis been performed appropriately and rigorously? 

Reviewer #1: Yes

Reviewer #3: No

Reviewer #4: I Don't Know

4. Have the authors made all data underlying the findings in their manuscript fully available?

Reviewer #1: Yes

Reviewer #3: Yes

Reviewer #4: Yes

5. Is the manuscript presented in an intelligible fashion and written in standard English?

Reviewer #1: Yes

Reviewer #3: Yes

Reviewer #4: No

6. Review Comments to the Author

Reviewer #1: thank you the Authors, all my comments are fully addressed including the sampling method or techniques and sample size.

Reviewer #3: Thank you for your interesting article and your hard work on that research.

I have few comments:

1- Title is long, Consider shortening the title for better impact (e.g., "Perceptions of Women and Providers on Barriers to Quality Emergency Obstetric Care in Oromia, Ethiopia").

2- There are frequent grammatical errors and awkward sentence constructions throughout the manuscript that compromise clarity. Examples:

• "An evidence showed..." should be "Evidence shows..."

• "Man powers" → "Human resources" or "Healthcare staff"

• "Gaps in lacks of training..." is redundant.

A thorough professional language edit is essential before resubmission.

3- In the methods:

- Why were these hospitals chosen? What criteria were used for saturation?

- Describe how data quality and validity were ensured beyond just "verbatim transcription."

4- MVA set in line 231: should be spelled out at first mention.

5- Only 5 out of the 32 references are up to date (within the last 5 years), could you please update some of your references?

Reviewer #4: ABSTRACT: there are some grammatical errors that make sentences difficult to understand. The 1st sentence in the methods section should be recasted because one can't conduct a study design (lines 28-29). The aim should be recasted ( lines (24-27). Abbreviations should 1st be written in full, then the abbreviation in bracket- this should be corrected throughout the text.

INTRODUC TION: There are spelling errors, incorrect tenses and grammatical errors. Lines 52-54 should be referenced, and the authors should explain how foetal distress is a cause of neonatal death. Several statements need to be referenced, some include lines 61-63, 65-67, 68-70, 73-75. The authors should explain how the arrived at their conclusion in line 70. Lines 59-61, 80-82 should be recasted. In relation to the context of rhe statements, the references used in lines 77-78 are rather old and should be substituted with more recent ones. The aim in lines 92-96 should be recasted for clarity.

METHODS. Due to grammatical errors, incorrect tenses, and poorly constructed sentences, several statements are difficult to understand eg, lines 143-145., lines 111=113, and 113-114 should be referenced, the authors should explain in line 116 how services were/are provided by integrated emergency obstetric surgery. They should explain in line 130, how integrated emergency obstetric surgery was an inclusion criterion and in lines 154-155 how the services are providers. In line 186 , why were verbal consents obtained from the participants and not written informed consents which is conventional. The authors should explain in much more detail how sample size was determined, how they selected their participants and interviewed them including the FGDs. It is not clear what the authors mean by pre-clapsia/eclapsia are they referring to pre-eclampsia/eclampsia.

RESULTS: The authors should correct the spelling of gynacology ( table 3)

7. PLOS authors have the option to publish the peer review history of their article (what does this mean?). If published, this will include your full peer review and any attached files.

Reviewer #1: No

Reviewer #3: No

Reviewer #4: No

---

## [Author Response · Author response to Decision Letter 2]

12 Dec 2025

Dear the manuscript reviewers and academic editor, first we appreciated your great input of our research work. Next, we incorporated all comments given from you for improving the manuscript for scientific community. Finally, we are still ready to take further comments on this manuscript if any

---

## [Decision Letter · Decision Letter 2]

12 Mar 2026

PONE-D-25-03007R2PERCEPTIONS OF WOMEN  AND PROVIDERS ON BARRIERS AND FACILITATORS OF QUALITY EMERGENCY OBSTETRIC CARE IN OROMIA, ETHIOPIAPLOS One

Dear Dr. Damme,

Thank you for submitting your manuscript to PLOS ONE. After careful consideration, we feel that it has merit but does not fully meet PLOS ONE’s publication criteria as it currently stands. Therefore, we invite you to submit a revised version of the manuscript that addresses the points raised during the review process.

**ACADEMIC EDITOR:** 

**Introduction**

Line 67-72:In the first paragraph, it would be important to incorporate the most recent evidence from the 2023 WHO report. Additionally, please ensure that the maternal mortality ratio (MMR) for Ethiopia is reported using the latest available data from the recent report to improve the accuracy and relevance of the background information.?Line 96-104: It is unclear why the discussion relies on the 2016 Ethiopian Demographic and Health Survey (EDHS) report despite the availability of a more recent report. Referring to the latest data would provide more up-to-date evidence and strengthen the relevance of the discussion.Line 105: you mentioned that “ The recent evidence shows that half of all maternal and one million newborn deaths can be prevented by providing high-quality care before, during and after childbirth. However, in low-income countries, obstetric mothers receive less than half of the recommended practices in atypical maternity care visit (14)”…how the study published in 2018 become recent ?The authors should clearly justify why a phenomenological approach was chosen to address the research questions. In addition, it would be helpful to explain why methodological triangulation was not considered. Alternatively, the authors should clarify whether this study is part of a broader or larger research project.Furthermore, the rationale for selecting the West Shoa zone as the study area should be clearly described. Since the study was conducted only in a limited geographic area, it would be more appropriate to specify West Shoa Zone in the title. Referring to the study area simply as Oromia Region may be misleading, as Oromia is a large region and findings from West Shoa alone may not adequately represent the entire region.
**Methods**
**Generally.**
**Reporting guidelines:** Please ensure you follow the appropriate reporting guidelines when preparing your manuscript and submit the completed checklist as supplementary material. Please state in the methods which guidelines were consulted when preparing the manuscript. More information on reporting guidelines can be found on the EQUATOR website (In this study, you can utilize **a SRQR checklist** to organize and report your technique section. Hence, you must adhere to this,  while revising your method sections )Check your study and source population …you mentioned only women ? what about health care workers ?
**Results**
You mentioned the themes “Barriers related to health care facilities,” “Barriers related to health care providers,” and “Barriers related to clients and the community as a whole.” However, the specific sub-themes under each theme should be clearly and explicitly presented so that readers can easily understand the structure and meaning of the findings. The same approach should also be applied to the facilitators, where the corresponding sub-themes should be clearly outlined under each main theme.The way the discussants’ statements are presented requires substantial improvement. When quoting health workers, please provide complete information in brackets, such as age, professional field, and years of experience. In addition, the language used should clearly reflect that the statements are directly expressed by the discussants, ensuring that the quotations accurately represent the participants’ voices. Thus, you have to take your time and work intensively on these sections.
**Discussion**
Generally, the discussion lacks depth, as it only compares the results across studies. When comparing the study findings with reports from the existing literature, it is recommended to comment on the reasons for the observed discrepancy based on the theoretical framework or give a plausible explanation. It is also required to indicate its clinical and theoretical implications on the topic of interest.
**Strengths and Limitations of Study**
Limitations should address potential biases (e.g., selection or recall bias), constraints in data collection (e.g., self-reporting or incomplete data), and generalizability of the findings to other populations or settings.
**General comments**
It needs English language editing, particularly grammar, comma  and tense use.3          Please based on your study design, stick to Research Reporting Checklist such as  **SRQR checklist**

We look forward to receiving your revised manuscript.

Kind regards,

Adera Debella Kebede, MSC

Academic Editor

PLOS One

Journal Requirements:

Additional Editor Comments:

Title : PERCEPTIONS OF WOMEN AND PROVIDERS ON BARRIERS AND FACILITATORS OF QUALITY EMERGENCY OBSTETRIC CARE IN OROMIA,ETHIOPIA

Introduction

1. Line 67-72:In the first paragraph, it would be important to incorporate the most recent evidence from the 2023 WHO report. Additionally, please ensure that the maternal mortality ratio (MMR) for Ethiopia is reported using the latest available data from the recent report to improve the accuracy and relevance of the background information.?

2. Line 96-104: It is unclear why the discussion relies on the 2016 Ethiopian Demographic and Health Survey (EDHS) report despite the availability of a more recent report. Referring to the latest data would provide more up-to-date evidence and strengthen the relevance of the discussion.

3. Line 105: you mentioned that “ The recent evidence shows that half of all maternal and one million newborn deaths can be prevented by providing high-quality care before, during and after childbirth. However, in low-income countries, obstetric mothers receive less than half of the recommended practices in atypical maternity care visit (14)”…how the study published in 2018 become recent ?

4. The authors should clearly justify why a phenomenological approach was chosen to address the research questions. In addition, it would be helpful to explain why methodological triangulation was not considered. Alternatively, the authors should clarify whether this study is part of a broader or larger research project.

5. Furthermore, the rationale for selecting the West Shoa zone as the study area should be clearly described. Since the study was conducted only in a limited geographic area, it would be more appropriate to specify West Shoa Zone in the title. Referring to the study area simply as Oromia Region may be misleading, as Oromia is a large region and findings from West Shoa alone may not adequately represent the entire region.

Methods

1. Generally. Reporting guidelines: Please ensure you follow the appropriate reporting guidelines when preparing your manuscript and submit the completed checklist as supplementary material. Please state in the methods which guidelines were consulted when preparing the manuscript. More information on reporting guidelines can be found on the EQUATOR website (In this study, you can utilize a SRQR checklist to organize and report your technique section. Hence, you must adhere to this, while revising your method sections )

2. Check your study and source population …you mentioned only women ? what about health care workers ?

Results

1 You mentioned the themes “Barriers related to health care facilities,” “Barriers related to health care providers,” and “Barriers related to clients and the community as a whole.” However, the specific sub-themes under each theme should be clearly and explicitly presented so that readers can easily understand the structure and meaning of the findings. The same approach should also be applied to the facilitators, where the corresponding sub-themes should be clearly outlined under each main theme.

2 The way the discussants’ statements are presented requires substantial improvement. When quoting health workers, please provide complete information in brackets, such as age, professional field, and years of experience. In addition, the language used should clearly reflect that the statements are directly expressed by the discussants, ensuring that the quotations accurately represent the participants’ voices. Thus, you have to take your time and work intensively on these sections.

Discussion

Generally, the discussion lacks depth, as it only compares the results across studies. When comparing the study findings with reports from the existing literature, it is recommended to comment on the reasons for the observed discrepancy based on the theoretical framework or give a plausible explanation. It is also required to indicate its clinical and theoretical implications on the topic of interest.

Strengths and Limitations of Study

Limitations should address potential biases (e.g., selection or recall bias), constraints in data collection (e.g., self-reporting or incomplete data), and generalizability of the findings to other populations or settings.

General comments

It needs English language editing, particularly grammar, comma and tense use.

1. Please based on your study design, stick to Research Reporting Checklist such as SRQR checklist and check your references  it is not as per journal require requirements.

Reviewers' comments:

Reviewer's Responses to Questions

**Comments to the Author**

1. If the authors have adequately addressed your comments raised in a previous round of review and you feel that this manuscript is now acceptable for publication, you may indicate that here to bypass the “Comments to the Author” section, enter your conflict of interest statement in the “Confidential to Editor” section, and submit your "Accept" recommendation.

Reviewer #1: All comments have been addressed

Reviewer #3: All comments have been addressed

Reviewer #4: (No Response)

2. Is the manuscript technically sound, and do the data support the conclusions?

Reviewer #1: Yes

Reviewer #3: Yes

Reviewer #4: Partly

3. Has the statistical analysis been performed appropriately and rigorously? 

Reviewer #1: Yes

Reviewer #3: Yes

Reviewer #4: I Don't Know

4. Have the authors made all data underlying the findings in their manuscript fully available?

Reviewer #1: Yes

Reviewer #3: Yes

Reviewer #4: Yes

5. Is the manuscript presented in an intelligible fashion and written in standard English?

Reviewer #1: Yes

Reviewer #3: Yes

Reviewer #4: No

6. Review Comments to the Author

Reviewer #1: (No Response)

Reviewer #3: Thank you for addressing the previous reviewers' comments. At this stage, I would suggest accepting this article for publication.

Reviewer #4: GENERAL: There are still several grammatical errors, poorly constructed sentences , incorrect tenses, incorrect use of plural and singular nouns and spelling errors that need to be addressed.

INTRODUCTION:Based on the context of the statement, the reference citing lines 58-60 should be a more recent one. Line 55 should better read -- Obstetric emergencies---. Lines 64-68-- Evidence shows ( and the statement should be referenced). Abbreviations should 1st be written in full, then the abbreviations in bracket eg EMONC, SDG. lines 78-79 SHOULD BE REFERENCED.

METHODS: The authors in lines 114-115 should explain why the 4 hospitals were selected. Several sentences should be recasted and grammar cor5rected eg, lines 119-120, 123-124, 132-134, 147-148, . Spelling errors in the inclusion criteria eg preeclapsia/eclapsia which correctly should be preeclampsia/eclampsia and this should be corrected throughout the manuscript. In lines 138=139, how emergency obstetric surgery an obstetric care provider? Abbreviations eg Gyn should 1st be written in full. In lines 154-155, the data instrument refereed to should be referenced.

RESULTS: Several grammatical and spelling errors eg lines 162, 189, 173, 176. Abbreviations in the tables should be written under them as footnotes. Lines 326-329, 364-367 should be recasted for clarity. In error in 335 should be corrected.

DISCUSSION: Several grammatical errors and wrong use of singular and plural nouns

REFERENCES: The referencing style should be uniform and in accordance with that of the journal. The following are either not complete, not well written or do not conform to the style of the journal, numbers 16, 22, 26, 27, 28, 30, 31, 33, 38.

7. PLOS authors have the option to publish the peer review history of their article (what does this mean?). If published, this will include your full peer review and any attached files.

Reviewer #1: No

Reviewer #3: No

Reviewer #4: No

---

## [Author Response · Author response to Decision Letter 3]

27 Mar 2026

Dear journal editor, I have approved for you that all the given comments were incorporate in the current edited manuscript

---

## [Decision Letter · Decision Letter 3]

4 May 2026

PONE-D-25-03007R3PERCEPTIONS AND EXPERIENCES OF WOMEN AND PROVIDERS ON BARRIERS AND FACILITATORS OF QUALITY EMERGENCY OBSTETRIC AND NEWBORN CARE SERVICES IN PUBLIC HOSPITALS OF WEST SHOA ZONE, OROMIA, ETHIOPIA: A PHENOMENOLOGICAL QUALITATIVE STUDYPLOS One

Dear Dr. Damme,

Thank you for submitting your manuscript to PLOS ONE. After careful consideration, we feel that it has merit but does not fully meet PLOS ONE’s publication criteria as it currently stands. Therefore, we invite you to submit a revised version of the manuscript that addresses the points raised during the review process.

You stated that “The Ethiopian demographic health survey (EDHS) in 2024-2025 also reported that the current Ethiopian maternal mortality rate and neonatal mortality rate were 169 deaths per 100,000 live births and25 deaths per 1000 live births, respectively” Are you sure that Ethiopian MMR is 169 ? kindly check figure again.

We look forward to receiving your revised manuscript.

Kind regards,

Adera Debella Kebede, MSC

Academic Editor

PLOS One

Journal Requirements:

Reviewers' comments:

Reviewer's Responses to Questions

**Comments to the Author**

1. If the authors have adequately addressed your comments raised in a previous round of review and you feel that this manuscript is now acceptable for publication, you may indicate that here to bypass the “Comments to the Author” section, enter your conflict of interest statement in the “Confidential to Editor” section, and submit your "Accept" recommendation.

Reviewer #4: All comments have been addressed

2. Is the manuscript technically sound, and do the data support the conclusions?

Reviewer #4: Yes

3. Has the statistical analysis been performed appropriately and rigorously? 

Reviewer #4: I Don't Know

4. Have the authors made all data underlying the findings in their manuscript fully available?

Reviewer #4: Yes

5. Is the manuscript presented in an intelligible fashion and written in standard English?

Reviewer #4: Yes

6. Review Comments to the Author

Reviewer #4: The authors have significantly improved the manuscript and it is now acceptable for publication. There are just few errors that need to be addressed editorially. Line 121- researcher should be researchers, line 310- carcity should be scarcity, line 3465 delivere - deliver. Then lines 479-484, 490-494 and 499-501 should be recasted

7. PLOS authors have the option to publish the peer review history of their article (what does this mean?). If published, this will include your full peer review and any attached files.

Reviewer #4: No

---

## [Author Response · Author response to Decision Letter 4]

6 May 2026

Thank you very much for your professional support to our manuscript and we incorporated all reviewer and editor comments in the manuscript

---

## [Editor Report · Decision Letter 4]

8 May 2026

PONE-D-25-03007R4PERCEPTIONS AND EXPERIENCES OF WOMEN AND PROVIDERS ON BARRIERS AND FACILITATORS OF QUALITY EMERGENCY OBSTETRIC AND NEWBORN CARE SERVICES IN PUBLIC HOSPITALS OF WEST SHOA ZONE, OROMIA, ETHIOPIA:A PHENOMENOLOGICAL QUALITATIVE STUDY

PLOS One

Dear Dr. Damme,

Thank you for submitting your manuscript to PLOS ONE. After careful consideration, we feel that it has merit but does not fully meet PLOS ONE’s publication criteria as it currently stands. Therefore, we invite you to submit a revised version of the manuscript that addresses the points raised during the review process.

You stated that “The Ethiopian Demographic and Health Survey (EDHS) 2024–2025 reported that the current maternal mortality rate and neonatal mortality rate in Ethiopia were 169 deaths per 100,000 live births and 25 deaths per 1,000 live births, respectively.” However, the figure of 169 deaths per 100,000 live births refers to the Pregnancy-Related Mortality Ratio (PRMR), not the Maternal Mortality Ratio (MMR). Standard reporting generally uses the Maternal Mortality Ratio; therefore, this figure should be checked and corrected accordingly.In addition, the citation format should follow Vancouver or modified Vancouver style. The current citation does not fully comply with this format and should therefore be revised and properly formatted.

We look forward to receiving your revised manuscript.

Kind regards,

Adera Debella Kebede, MSC

Academic Editor

PLOS One

Journal Requirements:

2.

---

## [Author Response · Author response to Decision Letter 5]

11 May 2026

First, we really thank your professional commitment for reviewing our manuscript. Additionally, we have incorporated all comments given and submitted the latest version of revised manuscript as the main article file for your consideration.

---

## [Editor Report · Decision Letter 5]

14 May 2026

PERCEPTIONS AND EXPERIENCES OF WOMEN AND PROVIDERS ON BARRIERS AND FACILITATORS OF QUALITY EMERGENCY OBSTETRIC AND NEWBORN CARE SERVICES IN PUBLIC HOSPITALS OF WEST SHOA ZONE, OROMIA, ETHIOPIA:A PHENOMENOLOGICAL QUALITATIVE STUDY

PONE-D-25-03007R5

Dear Dr.Gudissa Damme

We’re pleased to inform you that your manuscript has been judged scientifically suitable for publication and will be formally accepted for publication once it meets all outstanding technical requirements.

Kind regards,

Adera Debella Kebede, MSC

Academic Editor

PLOS One
---

## [Editor Report · Acceptance letter]

PONE-D-25-03007R5

PLOS One

Dear Dr. Damme,

I'm pleased to inform you that your manuscript has been deemed suitable for publication in PLOS One. Congratulations! Your manuscript is now being handed over to our production team.

Kind regards,

on behalf of

Dr. Adera Debella Kebede

Academic Editor

PLOS One